# Clinical Early-Onset Sepsis Is Equally Valid to Culture-Proven Sepsis in Predicting Outcome in Infants after Preterm Rupture of Membranes

**DOI:** 10.3390/jcm10194539

**Published:** 2021-09-30

**Authors:** Agnes Grill, Monika Olischar, Michael Weber, Lukas Unterasinger, Angelika Berger, Harald Leitich

**Affiliations:** 1Department of Pediatrics and Adolescent Medicine, Medical University of Vienna, 1090 Vienna, Austria; monika.olischar@meduniwien.ac.at (M.O.); lukas.unterasinger@meduniwien.ac.at (L.U.); angelika.berger@meduniwien.ac.at (A.B.); 2Department of Biomedical Imaging and Image-Guided Therapy, Medical University of Vienna, 1090 Vienna, Austria; michael.weber@meduniwien.ac.at; 3Department of Obstetrics and Gynecology, Medical University of Vienna, 1090 Vienna, Austria; harald.leitich@meduniwien.ac.at

**Keywords:** early-onset sepsis, mortality, preterm premature rupture of membranes, prematurity, severe morbidity

## Abstract

Background: Culture-proven sepsis is the gold standard in early-onset neonatal sepsis diagnosis. Infants born ≤29 weeks gestation after preterm rupture of membranes in the years 2009–2015 were included in a retrospective cohort study performed at a level III fetal-maternal unit. The study aimed to compare culture-proven sepsis, clinical sepsis and positive laboratory biomarkers ≤72 h as predictors of mortality before discharge and the combined outcome of mortality or severe short-term morbidity (severe cerebral morbidity, bronchopulmonary dysplasia and retinopathy). Results: Of the 354 patients included, culture-proven sepsis, clinical sepsis and laboratory biomarkers were positive in 2.3%, 8.5% and 9.6%, respectively. The mortality rate was 37.5% for patients with culture-proven sepsis (3/8), 33.3% for patients with clinical sepsis (10/30) and 8.8% for patients with positive laboratory biomarkers (3/34), respectively. Mortality or severe morbidity occurred in 75.0% of patients with culture-proven sepsis (6/8), 80.0% of patients with clinical sepsis (24/30) and 44.1% of patients with positive laboratory biomarkers (15/34), respectively. Conclusion: In preterm infants after preterm rupture of membranes, clinical sepsis was almost four times more common and at least equally valuable in predicting mortality and mortality or severe morbidity compared to culture-proven sepsis.

## 1. Introduction

Preterm premature rupture of membranes (pPROM), referring to membrane rupture before 37 weeks of gestation and before the onset of uterine contractions, occurs in 3% of pregnancies and is associated with one-third of all preterm births [1]. The underlying pathological process in pPROM includes intrauterine infection from ascending genital tract colonization. These infections may lead to increased cytokine activity with consequent enhancement of membrane apoptosis, production of proteases and dissolution of the membrane’s extracellular matrix [1]. Diagnosis is based on the patient’s medical history and a physical examination, and, in suspicious cases, on an immunochromatographic assay for detection of placental alpha microglobulin-1 protein [1]. PPROM implicates inflammation and infection and is a loss of barrier to infection ascending from the vagina [2]. Chorioamnionitis can be both a cause and a result of pPROM, and its highest observed occurrence is in the first week after pPROM [3]. Chorioamnionitis increases the likelihood of neonatal sepsis, which remains a major cause of neonatal mortality and morbidity in preterm infants [4]. The risk of sepsis increases with decreasing gestational age and birth weight [5]. In a prospective registry of very low birth weight infants in the United States, culture-proven early-onset sepsis rate accounts for 3.5% at gestational age <25 weeks and 1.9% in gestational ages 25 to 28 weeks, respectively [5]. However, the risk for sepsis increases in neonates born to mothers with chorioamnionitis. Sepsis rates of preterm infants born after pPROM vary from 2.5% to 36.1% [2,4,6,7,8,9]. The definition of early-onset sepsis after pPROM varies from blood culture-proven sepsis [4,6,9] to culture-proven sepsis in blood and cerebrospinal fluid [7,10], and culture-proven sepsis in any fluid [2,11] to clinical presentation consistent with sepsis without positive culture proof [2,9]. Further differences concern the time point of diagnosis, ranging from early onset in ≤72 h of life [7,9,11] to the first week of life [3] to any point during the initial hospitalisation [2,9]. The aim of the current study was to compare three different definitions of neonatal sepsis with regard to their predictive value on neonatal outcomes.

## 2. Materials and Methods

This was a retrospective, hospital-based cohort study carried out at a tertiary perinatal referral centre. Singleton and twin live inborn preterm patients ≤29 weeks gestation born after pPROM between 1 June 2009 and 31 December 2015 were included. Exclusion criteria were chromosomal abnormalities, major malformations or inborn errors of metabolism. pPROM was diagnosed by direct visualisation of amniotic fluid or by detecting insulin-like growth factor binding protein 1 in vaginal fluid. Standard management included antenatal corticosteroid treatment with betamethasone 12 mg, two doses 24 h apart and antibiotic therapy using ampicillin for 7 days [12]. Two tocolytic agents, atosiban or hexoprenaline were used in case of preterm labour, for a minimum of 48 h. Since 2015, magnesium sulfate was regularly administered in the last 24 h before delivery in all pregnancies <32 gestational weeks for fetal neuroprotection. Obstetric and perinatal factors were extracted from case records. The following obstetric factors were assessed: maternal age, occurrence of anhydramnios (defined as largest vertical amniotic fluid pocket less than 2 cm, as determined by abdominal ultrasound), time interval between pPROM and birth, use of antibiotics, tocolytics and antenatal steroids. Maternal laboratory test results were defined as positive for infection when maternal white blood cell count > 16,000/nL and C-reactive protein (CRP) level > 5 mg/dL were present within the last three days prior to birth. Clinical chorioamnionitis was suspected in the presence of maternal fever of more than 38 °C, maternal tachycardia > 100 beats/min, purulent vaginal discharge, uterine tenderness and, or fetal tachycardia > 160 beats/min. Mode of delivery and cord blood analysis were registered. Microbiological cultures of placental tissue and the amniotic membrane for bacterial and fungal growth were assessed as part of the clinical routine after pPROM. For all study patients, the following perinatal data were assessed: gestational age at birth, gender, birth weight, APGAR-Score at 5 min, pH and serum lactate level of the infant’s venous blood gas analysis in the first hour of life. Standard delivery room management consisted of early high-flow continuous positive airway pressure (CPAP) and application of surfactant via thin catheter in spontaneously breathing infants ≤27 weeks gestation as prophylactic therapy and ≤29 weeks gestation as early rescue therapy, using less invasive surfactant administration (LISA) [13]. Blood cultures were obtained [14] and broad-spectrum antibiotics (ampicillin and gentamicin) were administered in all infants ≤27 weeks gestation after birth. Antibiotics were discontinued at 48 h if blood cultures obtained at birth and laboratory biomarkers of infection remained negative. Blood sampling was done on day 1 and 3 routinely. Neonatal laboratory biomarker results ≤72 h of life were defined positive for infection when two or more of the following results were positive: CRP ≥ 2 mg/dL, Interleukin 8 (IL-8) > 100 pg/uL, Interleukin 6 (IL-6) > 100 pg/mL, ratio of immature to total neutrophil counts (I/T-ratio) ≥ 0.2, white blood cell count <4000 or > 20.000 G/L. Routine measurement of IL-8 was discontinued in 2015 and replaced by IL-6. We registered results of blood cultures ≤72 h of life. Clinical symptoms consistent with early onset neonatal sepsis were defined as respiratory, cardiovascular or feeding dysfunction beyond the normal range for each gestational week diagnosed ≤72 h of life. These symptoms included respiratory dysfunction with severe apnea and need for therapy step-up (caffeine, doxapram, bi-level CPAP, non-invasive ventilation) or mechanical ventilation or cardiovascular dysfunction with inotropic support or temperature instability, tachycardia, prolonged capillary refill time or severe feeding problems with abdominal distention and total parenteral nutrition. The diagnosis of clinical sepsis was made by the attending NICU team if at least one of the symptoms described above was present. Mortality was defined as mortality before discharge from hospital. Bronchopulmonary dysplasia (BPD) was defined as oxygen dependency at 36 weeks postmenstrual age. Severe cerebral morbidity was present if either grade III intraventricular haemorrhage (IVH) or grade III IVH with periventricular haemorrhagic infarction or cystic periventricular leukomalacia (c-PVL grade II, III, IV) was diagnosed [15,16]. Incidence of severe retinopathy (≥grade 3) of prematurity (ROP) was assessed. Severe morbidity was defined as a composite of severe cerebral morbidity, BPD and/or ROP. The following three non-overlapping groups of different definitions of neonatal sepsis at ≤72 h of life were evaluated for their value in predicting mortality and the combined outcome of mortality or severe morbidity: group one was culture-positive early-onset sepsis, defined as a positive result from the blood culture taken immediately after birth plus clinical signs of sepsis, as defined above; group two was clinical early-onset sepsis, defined as positive laboratory biomarkers plus clinical symptoms consistent with neonatal sepsis plus antibiotic therapy ≥ 5 days with negative blood-culture results; and group three included patients with positive laboratory biomarkers plus antibiotic therapy ≥ 5 days with negative blood culture and no clinical symptoms consistent with sepsis.

Statistical analysis: Data were analyzed with the IBM SPSS statistical package for Windows version 23.0 (IBM, Armonk, NY, USA). Nominal data are presented using percentages as well as absolute numbers. Metric data are presented using mean ± standard deviation, as well as median, minimal and maximal values. Crosstab and Pearson Chi-Square test or Fisher’s Exact test were used to compare nominal data. For metric data, a Student’s unpaired *t*-test (in normally distributed data which was tested using the Kolmogorov–Smirnov test) or the Mann–Whitney *U* test (in the case of skewed data).

## 3. Results

In our study, 363 patients born ≤29 weeks of gestation after pPROM met the inclusion criteria. Due to missing data of blood cultures or laboratory test results, nine patients were excluded. Demographic data of maternal and patient characteristics for the whole cohort are presented in Table 1. In our study population, a high rate of antenatal steroids (99.4%) and caesarean section (89.2%) was found. Maternal laboratory biomarkers for infection were positive in 4.3% of cases. Clinical chorioamnionitis was present in only 1.7%, whereas placental bacterial culture was positive in 54.3% of cases. Mean latency period between pPROM and birth was 7 ± 13 days. Mean GA at pPROM was 25.7 ± 2.7 weeks, mean GA at birth 26.7 ± 1.9 and mean birthweight 924 ± 255 g. Characteristics of the study groups are presented in Table 2. Comparing the three groups, there were no significant differences concerning GA at pPROM, GA at birth and birth weight. Of the parameters reflecting postnatal adaptation, only venous pH in the first hour of life was significantly different between the three groups.

Culture-proven sepsis was diagnosed in 2.3% of patients (8/354). Blood cultures were positive for Escherichia coli in five cases and for Haemophilus influenzae, Morganella morganii and Candida glabrata in one case each. Clinical sepsis was diagnosed in 8.5% of patients (30/354). Neonatal laboratory biomarkers were positive for infection without clinical symptoms consistent with sepsis in 9.6% of patients (34/354). The overall mortality rate in the whole cohort was 11.8% and the rate for mortality or severe morbidity was 33.3%. Mortality rate was 37.5% for patients with culture-proven sepsis (3/8), 33.3% for patients with clinical sepsis (10/30) and 8.8% for patients with positive laboratory biomarkers (3/34), respectively (*p* = 0.034, Table 3). Mortality or severe morbidity occurred in 75.0% of patients with culture-proven sepsis (6/8), 80.0% of patients with clinical sepsis (24/30) and 44.1% of patients with positive laboratory biomarkers (15/34), respectively (*p* = 0.009). Severe morbidities were presented in detail in Table 3. Regarding the panel of neonatal laboratory biomarker results ≤72 h of life CRP, IL-8 and I/T ratio correlated significantly with mortality as well as mortality or severe morbidity. IL-6 and maximum white blood cell count showed a significant correlation with mortality or severe morbidity (Table 4). There was a trend towards a higher rate of laboratory biomarker positive for infection in patients delivered after labour unresponsive to tocolysis (26.3% versus 18.3%; *p* = 0.095).

## 4. Discussion

Our results indicate that clinical sepsis was almost four times more common than culture-proven sepsis in preterm infants ≤29 weeks of gestation born after pPROM. The diagnosis of clinical sepsis was at least equally valuable to predict mortality and mortality or severe morbidity compared to the gold standard of culture-positive sepsis in this patient cohort.

The gold standard of neonatal sepsis is the isolation of a pathogen from a blood culture, as this confirms the diagnosis of bacteraemia and allows for identification and susceptibility testing on the organism to optimise choice and duration of antimicrobial therapy [14]. Blood cultures, though, may lead to false-negative results, and despite sterile blood cultures, a clinical course consistent with sepsis is possible [17]. The generally low rate of culture-proven sepsis may underestimate the clinical problem of early-onset sepsis in preterm infants.

The sensitivity of blood cultures to detect neonatal bacteraemia depends on the number of cultures obtained and the amount and availability of samples [17]. The intended amount of 0.5–1 mL of blood per sample is sometimes difficult to obtain immediately after birth before initiating antimicrobial therapy, especially in very sick and very small preterm infants. A high percentage of antepartum antibiotic use—98.8% in our cohort—may also negatively impact the sensitivity of blood cultures [17]. Since time to culture positivity exceeds 48 h in less than 2.0% [10] of samples, empiric antibiotics can be discontinued at least after 48 h when cultures remain negative.

Definitions of neonatal sepsis and reported sepsis rates after pPROM vary. Culture-proven sepsis is the most commonly used definition [4,6,7]. Clinical presentation consistent with sepsis was used by two study groups [2,9] and only one of these groups [9] included neonatal laboratory signs for clinical diagnosis of neonatal sepsis.

In our study, several laboratory parameters as stand-alone factors were significantly related to neonatal outcomes. The majority of these biomarkers are components of the inflammatory cascade, and no single biomarker can fulfil all the criteria of an ideal biomarker [18]. Early biomarkers, such as IL-6, were found to be significantly upregulated as early as two days before clinical diagnosis of sepsis [19]. A bedside IL-6 test, however, was less sensitive for identifying early-onset as compared to late-onset sepsis cases [20]. An in comparison late biomarker, such as CRP, at 24 and 48 h after the initial presentation can accurately confirm the infection and allow discontinuation of antibiotics in non-sepsis cases [18]. All these mediators will respond to any event triggering an inflammatory host reaction, as seen in our analysis in the trend towards a higher rate of positive biomarkers after spontaneous labour, so clinical judgement is vital for effective interpretation of biomarker results [18]. In our study, the predictive power of laboratory biomarkers alone was inferior compared to the definition of clinical sepsis, including clinical parameters of infection.

A limitation of our study was its retrospective design. The strength of our study was the high number of extremely preterm infants due to effective centralisation.

We conclude that a strict definition of clinical sepsis, including positive clinical and laboratory parameters of infection, was at least equally valuable to culture-proven sepsis in predicting mortality and mortality or severe morbidity in our cohort of extremely preterm infants ≤29 weeks of gestation. We propose that clinical sepsis, strictly defined, as in the current study, in addition to culture-proven sepsis, rather than the exclusive consideration of blood culture results, better reflects the true rate of infected preterm infants with adverse prognosis after pPROM.

## Figures and Tables

**Table 1 jcm-10-04539-t001:** Maternal and patient characteristics of the whole study population (*n* = 363).

Maternal Age (yrs ± SD)	32 ± 5.9
Antibiotics (%)	98.8
Tocolytics (%)	99.4
Labour unresponsive to tocolysis (%)	26.2
Antenatal steroids (%)	99.4
Caesarean section (%)	89.2
Anhydramnios (%)	13.4
Maternal laboratory biomarkers positive for infection (%)	4.3
Clinical chorioamnionitis (%)	1.7
Positive placental culture (%)	54.3
Latency time pPROM-birth (days), mean ± SD, median (min–max)	7 ± 13, 3 (0–94)
GA at pPROM (weeks, mean ± SD)	25.7 ± 2.7
GA at birth (weeks, mean ± SD)	26.7 ± 1.9
Gender (male, %)	58.7
Birth weight (g, mean ± SD)	924 ± 255

pPROM: preterm premature rupture of membranes; GA: gestational age.

**Table 2 jcm-10-04539-t002:** Patient characteristics of the three groups of infants with sepsis (*n* = 72).

	Culture-Proven Sepsis (*n* = 8)	Clinical Sepsis (*n* = 30)	Positive Laboratory Biomarkers (*n* = 34)	*p* Value
GA at pPROM (weeks, mean ± SD)	24.6 ± 1.9	23.5 ± 2.3	24.9 ± 2.0	0.060
GA at birth (weeks, mean ± SD)	25.4 ± 1.2	25.1 ± 1.6	25.7 ± 1.5	0.372
Gender (male, %)	50.0	56.7	55.9	1.000
Birth weight (g, mean ± SD)	801 ± 99	724 ± 206	815 ± 206	0.190
APGAR 5 min < 7(%)	28.6	20.0	8.8	0.227
Cord blood pH (mean ± SD)	7.27 ± 0.24	7.29 ± 0.10	7.34 ± 0.07	0.120
Venous pH (mean ± SD)	7.03 ± 0.15	7.14 ± 0.11	7.19 ± 0.11	0.034
Venous lactate (mg/dL, mean ± SD)	107 ± 83	35 ± 21	38 ± 28	0.415

GA: gestational age; pPROM: preterm premature rupture of membranes.

**Table 3 jcm-10-04539-t003:** Mortality, mortality or severe morbidity and the severe morbidities in detail in relation to the three different definitions of neonatal sepsis.

	Mortality	Mortality or Severe Morbidity	Severe IVH	c-PVL	Severe ROP	BPD
Culture-proven sepsis (8/354)	37.5% (3/8)	75.0% (6/8)	25.0% (2/8)	12.5% (1/8)	0.0% (0/8)	25.0% (2/8)
Clinical sepsis (30/354)	33.3% (10/30)	80.0% (24/30)	30.0% (9/30)	0.0% (0/30)	36.7% (11/30)	36.7% (11/30)
Positive laboratory biomarkers (34/354)	8.8% (3/34)	44.1% (15/34)	11.8% (4/34)	0.0% (0/34)	26.5% (9/34)	20.6% (7/34)

Severe IVH: grade III intraventricular haemorrhage or grade III IVH with periventricular haemorrhagic infarction; c-PVL: cystic periventricular leukomalacia; severe ROP: retinopathy of prematurity ≥ grade 3; BPD: bronchopulmonary dysplasia.

**Table 4 jcm-10-04539-t004:** Correlation of mortality and mortality or severe morbidity with neonatal laboratory test results ≤72 h of life.

	Total	Mortality		Mortality or Severe Morbidity	
		yes	no	yes	no
Max. CRP mg/dl, mean ± SD, median (min–max)	*n* = 359	2.1 ± 2.2 * 1.3 (0.2–8.8)	1.2 ± 1.3 * 0.8 (0.0–12.1)	1.7 ± 1.9 * 1.1 (0.0–12.1)	1.1 ± 1.1 * 0.8 (0.0–8.0)
Max. IL-8 pg/ul, mean ± SD, median (min-max)	*n* = 335	1017 ± 2899 * 91 (29–15.000)	164 ± 623 * 65 (6–8964)	455 ± 1701 * 83 (20–15.000)	157 ± 697 * 59 (6–8964)
Max. IL-6 pg/mL, mean ± SD, median (min-max)	*n* = 76	86 ± 36 92 (35–122)	123 ± 241 33 (2–1262)	239 ± 345 * 92 (5–1262)	86 ± 179 * 21 (2–970)
I/T ratio, mean ± SD, median (min-max)	*n* = 312	0.19 ± 0.13 * 0.16 (0.04–0.62)	0.11 ± 0.10 * 0.08 (0.00–0.67)	0.17 ± 0.12 * 0.14 (0.01–0.62)	0.1 ± 0.10 * 0.07 (0.00–0.67)
Min. white blood cell count G/L, mean ± SD, median (min-max)	*n* = 333	11.500 ± 9970 8080 (2210–45.570)	11.850 ± 8220 9130 (2080–67.650)	12.880 ± 9220 9660 (2210–45.570)	11.310 ± 7970 8940 (2080–67.650)
Max. white blood cell count G/L, mean ± SD, median (min-max)	*n* = 334	19.260 ± 16.660 14.830 (4240–70.270)	15.810 ± 10.650 12.320 (4030–74.070)	18.860 ± 12.130 * 15.000 (4240–70.270)	14.910 ± 10.500 * 11.870 (4030–74.070)

* *p* < 0.05; CRP: C-reactive protein; IL: interleukin; I/T ratio ratio of immature to total neutrophils.

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
