# Peer review of "Clinical Early-Onset Sepsis Is Equally Valid to Culture-Proven Sepsis in Predicting Outcome in Infants after Preterm Rupture of Membranes"

_jcm, 2021, doi:10.3390/jcm10194539_

Round 1
Reviewer 1 Report
Dear Sir:
The manuscript entitled “Clinical early-onset sepsis is equally valid to culture-proven sepsis in predicting outcome in infants after preterm rupture of membranes, Stage 1” has been reviewed. The results are interesting and reasonable. However, there are some points that require clarification.
- About the PPROM, was there any difference about the duration of the condition before delivery? Did all included mothers complete the 7-day antibiotics treatment? If not, was there any influence to patient outcomes among different maternal PPROM duration and antibiotics treatment duration?
- Could authors provide more specific description about the symptoms/signs and onset time/duration about clinical sepsis? Were they diagnosed by one physician or under certain criteria? The description “more than expected” is not specific enough. Maybe a table can help.
- Did all three groups of patients receive the same antibiotics and same treatment duration? Were there any criteria about the duration of antibiotics treatment more than 5 days? How about the dosage of antibiotics?
- About the biomarkers, “two or more of the biomarkers were positive” is not specific. Could authors provide some tables about the percentage of positive marker? The timing of checking the biomarkers is not clear. Some description about the timing of taking blood sample is necessary.
- About the outcomes (mortality and morbidity), since most of the patients were extreme prematurity, how to be sure about the effects of other possible factors, such as recurrent sepsis or ventilator associated pneumonia? The timing of the occurrence or diagnosis should be mentioned.
- About Table 1, since the description of the conditions are about the mother, the title of table is suggested to remove the words “and patient”.
Author Response
Dear reviewer,
thank you for giving us the opportunity to present our revised paper.
- All mothers finished a 7 day course of antibiotics as our treatment standard except those who delivered before. Data on latency between pPROM and birth are presented in table 1. Evaluation of latency between pPROM and birth was not the topic of this study, however in a further study we adressed this issue without finding any significant effect on outcome. (Is it possible to make a reliable prognosis within the first hour of life for very low birth weight infants delivered after preterm premature rupture of membranes? Messerschmidt A, Olischar M, Birnbacher R, Sauer A, Weber M, Puschnig D, Unterasinger L, Pollak A, Leitich H. Neonatology. 2011;99(2):146-52. doi: 10.1159/000313969. Epub 2010 Aug 27.)
2.1. The definition of clinical sepsis was revised.
2.2. The onset was defined as <72 hours of life.
2.3. Diagnosis of clinical sepsis was made by the attending NICU team.
2.4. The description „more than expected“ was replaced.
The following sentences were added to the paper: Clinical symptoms consistent with early onset neonatal sepsis were defined as respiratory, cardiovascular or feeding dysfunction beyond the normal range for each gestational week diagnosed ≤72 hours of life. These symptoms included respiratory dysfunction with severe apnea and need for therapy step-up (caffeine, doxapram, bi-level CPAP, non-invasive ventilation) or mechanical ventilation or cardiovascular dysfunction with inotropic support or temperature instability, tachycardia, prolonged capillary refill time or severe feeding problems with abdominal distention and total parenteral nutrition. The diagnosis of clinical sepsis was made by the attending NICU team if at least one of the symptoms described above was present.
- All three groups received the full 7 day course of antibiotics. Criteria for a full course were positive neonatal laboratory biomarkers (as defined in the study) <72 hours of life. Treatment consisted of 5 days gentamicin (day 1 5mg/kg, day 2-5 3.5mg/kg) and 7 days ampicillin (150mg/kg).
4.1. As described in the method section positive neonatal laboratory biomarker were defined positive for infection when two or more of the following results were positive: CRP ≥2 mg/dl, Interleukin 8 (IL-8) >100 pg/ul, Interleukin 6 (IL-6) >100 pg/ml, ratio of immature to total neutrophil counts (I/T-ratio) ≥0,2, white blood cell count <4.000 or >20.000 G/L.
4.2. We are sorry, but because the period of time to revise the paper is very short (5 days) we are not able to provide additional analyses of our data set. Correlation of neonatal laboratory test results ≤72 hours of life with mortality and mortality or severe morbidity is presented in table 4.
4.3. Blood sampling was done on day 1 and 3 routinely. This was added to the text.
- As the data set focussed on early onset sepsis <72 hours of life information on late onset sepsis and ventilator asscoiated pneumonia were not available in our data set. Diagnosis of short term morbidities were extracted at the end of the hospital stay.
- At the end of table 1 neonatal data such as GA at birth, gender and birth weight were presented for the whole study population. Therefore „and patients“ was left in the titel.
Reviewer 2 Report
The presented study tackles an issue of Predicting Outcome in Preterm Born Infants after Preterm Rupture of Membranes. The study was conducted reliably with appropriate selection of tests.
However some issues require complementary information:
- I suggest changing the title- eg. Predicting Outcome in Preterm Born Infants after Preterm Rupture of Membranes- Preliminary Study.
- Verse 12-13- I suggest deleting the sentence starting from „Clinical nad laboratory…”
- Verse 13-15- I suggest reediting that sentence eg. „ in the years 2009-2015” and „ The study group included infants borned…..”
- Introduction- I suggest including the information about etiopathology of pPROM and diagnostic methods
- Verse 56- I suggest changing „PPROM” on „pPROM”
- I suggest including more information in results section- The tables should be elaborated
- I suggest including the information about the numer of positive bacterial cultures from placenta and comapring it with the groups
Author Response
Dear reviewer,
thank you for giving us the opportunity to present our revised paper.
- The proposed title is too unspecific because it suggests a general risk factor analysis. In our study we compared 3 different models of diagnosing early onset neonatal sepsis and their predictive value for outcome.
- Verse 12-13: The sentence was deleted.
- Verse 13-15 was reedited.
4.1. Information about etiopathology of pPROM was added.
The following sentence was added: The underlying pathologic process in pPROM includes intrauterine infection from ascending genital tract colonization. These infections may lead to increased cytokine activity with consequent enhancement of membrane apoptosis, production of proteases, and dissolution of the membrane’s extracellular matrix.
4.2. Information on diagnosis was added.
The following sentence was added: Diagnosis is done based on history and physical examination, and in suspicious cases by immunochromatographic assay for detection of placental alpha microglobulin-1 protein.
- Verse 56- „PPROM” was changed to „pPROM”
- More information from the tables was included in the result section.
In our study population a high rate of antenatal steroids (99.4%) and caesarean section (89.2%) was found. Maternal laboratory biomarkers for infection were positive in 4.3% of cases. Clinical chorioamnionitis was present in only 1.7%, whereas placental bacterial culture was positive in 54.3% of cases. Mean latency period between pPROM and birth was 7 ± 13 days. Mean GA at pPROM was 25.7 ± 2.7 weeks, mean GA at birth 26.7 ± 1.9, and mean birthweight 924 ± 255g.
Comparing the three groups there were no significant differences concerning GA at pPROM, GA at birth, and birth weight. From the parameters reflecting postnatal adaptation only venous pH in the first hour of life was significantly different between the 3 groups.
- We are sorry, but because the period of time to revise the paper is very short (5 days) we are not able to provide additional analyses of our data set.